# A Computational Approach Applied to the Study of Potential Allosteric Inhibitors Protease NS2B/NS3 from Dengue Virus

**DOI:** 10.3390/molecules27134118

**Published:** 2022-06-27

**Authors:** Renato A. da Costa, João A. P. da Rocha, Alan S. Pinheiro, Andréia do S. S. da Costa, Elaine C. M. da Rocha, Rai. C. Silva, Arlan da S. Gonçalves, Cleydson B. R. Santos, Davi do S. B. Brasil

**Affiliations:** 1Graduate Program in Science and Environment, Institute of Exact and Natural Sciences, Federal University of Pará (UFPA), Belém 66075-110, PA, Brazil; alansp@ufpa.br (A.S.P.); ac165051@gmail.com (A.d.S.S.d.C.); davibb@ufpa.br (D.d.S.B.B.); 2Federal Institute of Education, Science and Technology of Pará Campus Castanhal, Castanhal 68740-970, PA, Brazil; 3Federal Institute of Education, Science and Technology of Pará—Campus Bragança, Bragança 68600-000, PA, Brazil; joao.rocha@ifpa.edu.br (J.A.P.d.R.); elaine.rocha@ifpa.edu.br (E.C.M.d.R.); 4Departamento de Química, Faculdade de Filosofia, Ciências e Letras de Ribeirão Preto, Universidade de São Paulo, Ribeirão Preto 14040-901, SP, Brazil; raics@usp.br; 5Federal Institute of Education, Science and Technology of Espírito Santo, Vila Velha 29106-010, ES, Brazil; arlangoncalves@gmail.com; 6Laboratory of Modeling and Computational Chemistry, Department of Biological and Health Sciences, Federal University of Amapá, Macapá 68902-280, AP, Brazil; breno@unifap.br

**Keywords:** Dengue, NS2B/NS3pro, allosteric site, molecular docking, molecular dynamics and free energy

## Abstract

Dengue virus (DENV) is a danger to more than 400 million people in the world, and there is no specific treatment. Thus, there is an urgent need to develop an effective method to combat this pathology. NS2B/NS3 protease is an important biological target due it being necessary for viral replication and the fact that it promotes the spread of the infection. Thus, this study aimed to design DENV NS2B/NS3pro allosteric inhibitors from a matrix compound. The search was conducted using the Swiss Similarity tool. The compounds were subjected to molecular docking calculations, molecular dynamics simulations (MD) and free energy calculations. The molecular docking results showed that two compounds, ZINC000001680989 and ZINC000001679427, were promising and performed important hydrogen interactions with the Asn152, Leu149 and Ala164 residues, showing the same interactions obtained in the literature. In the MD, the results indicated that five residues, Lys74, Leu76, Asn152, Leu149 and Ala166, contribute to the stability of the ligand at the allosteric site for all of the simulated systems. Hydrophobic, electrostatic and van der Waals interactions had significant effects on binding affinity. Physicochemical properties, lipophilicity, water solubility, pharmacokinetics, druglikeness and medicinal chemistry were evaluated for four compounds that were more promising, showed negative indices for the potential penetration of the Blood Brain Barrier and expressed high human intestinal absorption, indicating a low risk of central nervous system depression or drowsiness as the the side effects. The compound ZINC000006694490 exhibited an alert with a plausible level of toxicity for the purine base chemical moiety, indicating hepatotoxicity and chromosome damage in vivo in mouse, rat and human organisms. All of the compounds selected in this study showed a synthetic accessibility (SA) score lower than 4, suggesting the ease of new syntheses. The results corroborate with other studies in the literature, and the computational approach used here can contribute to the discovery of new and potent anti-dengue agents.

## 1. Introduction

Dengue flavivirus (DENV) is a virus transmitted by mosquitoes and occurs in more than 100 countries. In the vast majority of cases, DENV presents in humans as a mild disease characterized by fever, myalgia, arthralgia and retroocular headache, which improve without intervention or any complication; however, severe forms can develop due to plasma leakage, which can quickly be fatal. DENV is a tropical disease; its highest amount of growth has occurred over the last three decades. DENV has an annual incidence of approximately 50 million cases. Worldwide, it has threatened over 50% of the population. Specifically, its mortality rate in the Americas is approximately 1.2%. Therefore, in the absence of an effective vaccine or therapeutic approach, there is an urgent need to develop new therapies [1,2,3].

Dengue virus has an 11 kb positive-sense single-stranded genomic RNA encoding a precursor polyprotein (5′-C-prME-NS1-NS2A-NS2B-NS3-NS4A-NS4BNS5-3′) that is cleaved to produce three structural proteins (capsid protein C, membrane protein prM and envelope protein E) and seven nonstructural proteins (NS1, NS2a, NS2b, NS3, NS4a, NS4b and NS5). Of these, the NS2B/NS3 protease complex inhibits the host type I interferon (IFN) pathway by cleaving a key adaptor protein, the human mediator of IRF3 activation (MITA). The NS2B/NS3 protease is thus required for viral replication and promotes the progression of DENV infection [4,5,6,7]. Therefore, it has been selected as an important target for the design of drugs against dengue infection [8,9].

A promising alternative is the design of small molecules directed to the allosteric site. Allosteric sites are defined as regions of the protein that, when linked to a small-molecule ligand, undergo a change in conformation or an alteration in their conformational balance that affects enzyme function. In the literature, several articles are found exploring the allosteric sites of DENV NS2B/NS3pro in search of more potent inhibitors [10,11,12,13,14,15]. In DENV NS2B/NS3pro, this binding site is located behind the active site and is formed mainly by the residues NS3-Asp71, NS3-Lys73, NS3-Lys74, NS3-Trp83, NS3-Leu85, NS3-Gly87, NS3-Glu88, NS3-Trp89, NS3- Glu91, NS3-Thr118, NS3-Thr120, NS3-Val147, NS3-Leu149, NS3-Asn152, NS3-Val155, NS3-Ala164, NS3-Ile165 and NS3-Asn167 [12].

The fact that the DENV NS3pro active site is planar and preferentially binds to peptide-based substrates, such as the competitive peptidomimetics inhibitors of NS2B/NS3pro DENV found in the literature, has conferred limited oral bioavailability and difficulties in in vivo assays in studies of potential inhibitors targeting the active site of this enzyme [13,16,17]. However, this strengthens the strategy of exploring allosteric sites that are known to be successful targets of small molecule inhibitors, as presented in the works of [13,14,15,16].

Because of the need to develop more effective therapeutic agents and insurance against DENV, the aim of this work is to carry out the virtual screening of inhibitor-like compounds, analogous to tiaguanine, proposed by Hariono et al. [18]. In addition, we plan to target these compounds to the allosteric site in order to design DENV NS2B/NS3pro non-competitive inhibitors. Thus, we performed molecular docking, molecular dynamic simulations and free-energy calculations. We hope to contribute to the efforts in discovering novel and potent anti-DENV agents.

## 2. Results and Discussion

### 2.1. Molecular Docking

In this work, 71 ligands were selected and submitted to molecular docking simulations where the ranking was performed according to the binding affinity energy values obtained by the PLPchem scoring function of the GOLD program. The four best ligands (highest energy) were selected to assess the interactions between the residues of the NS2B/NS3pro (PDB 2FOM) allosteric site and the ligand (Table 1).

From the docking results, several interactions were identified regarding the residues of the NS2B/NS3pro allosteric site with ZINC000001680989, ZINC000001679427, ZINC000006694490 and ZINC000001684485. For the ligand ZINC000001680989, hydrogen bond interactions were observed with the Leu179(149), Asn182(152) and Ala194(164) residues, and hydrophobic interactions were observed with the Leu104(74), Leu106(76) and Trp113(83) residues; the latter also resulted in π–π interactions. For ZINC000001679427, hydrogen bond interactions with the residues Lys104(74), Leu179(149) and Ala194(164) were observed; hydrophobic contacts were also reported with the residues Trp113(83), Gly178(148), Val184(154) and Ala194(164).

For ZINC000006694490, hydrogen bond interactions with the residues Asn182(152), Thr150(120) and Val185(155)—along with hydrophobic interactions with Leu106(76) and Val184(154)—were observed. ZINC000001684485 interacted with hydrogen bonds in Thr150(120), Asn182(152) and Asn197(167), and hydrophobic interactions were observed with Val184(154) and Thr150(120). These interactions are shown in Figure 1. The numbers in the parentheses indicate the numbering within the Erbel model [19].

Notably, all of the ligands interacted with residues within the DENV NS2BNS3pro allosteric cavity [12]. The ZINC000001680989, ZINC000001679427 and ZINC000001684485 binders interacted with the Asn182(152) residue, which is considered essential for the interaction with NS2B/NS3pro allosteric inhibitors and is in accordance with other molecular coupling studies that involve DENV NS2B/NS3pro allosteric inhibitors [12,13,20]. This suggests that these ligands may show inhibitory activity toward the NS2B/NS3pro allosteric site. In particular, it was observed that the ligands ZINC000001680989 and ZINC000001679427 formed hydrogen bond interactions with the residues Asn182(152), Leu179(149) and Ala194(164), which is in agreement with the results reported by Mukhametov et al. [12] and Millies et al. [13] for DENV NS2B/NS3pro allosteric inhibitors in their studies in silico and in vitro. Another fact that corroborated this result is that these binders presented the best PLPChem score energies according to the PLPchem scoring function.

### 2.2. Molecular Dynamics

To prove the stability and conformational changes, MD simulation was performed to understand the characteristics of the NS2B/NS3pro-ZINC000001680989, NS2B/NS3pro-ZINC000001679427, NS2B/NS3pro-ZINC000006694490 and NS2B/NS3pro-ZINC000001684485 ratios in relation to the nanosecond scale. According to Figure 2, the backbone RMSDs for the NS2B/NS3pro-ligand complexes showed that the selected ligands remained stable at the protein allosteric site, converging at an RMSD value below 2 Å, while in the native NS2B/NS3 protein, the RMSD fluctuation is significantly greater, suggesting the inhibitory effect of the ligands used in this work. Fluctuations between 1 and 3 Å within a reference protein structure are perfectly acceptable and reflect complex stability [21].

For a more refined visualization, the average RMSD values were extracted with their error bars. The results suggested that the protein/ZINC0000016799427 complex had a better stabilization in relation to the other complexes (Figure 3).

In addition, an analysis of Figure 3 showed that the RMSD values of the NS2B/NS3pro-ZINC000001679427 and NS2B/NS3pro-ZINC000001680989 complexes fluctuated within a range of ±1.2 Å over time, thus exhibiting the greater stability of the protein complex link during the simulation [22]. Notably, these binders presented the best PLPChem score and interacted with the essential residues for inhibition at the DENV NS2B/NS3 allosteric site [12,13,16].

Local fluctuations in the protein are measured by the mean quadratic fluctuation (RMSF) and are useful for characterizing local changes along the protein chain [23]. The RMSF values of NS2B/NS3pro after binding to the four ligands are shown in Figure 4. The peaks in the graph indicate the amino acid residues in the protein that fluctuate the most during the MD simulation. Typically, the tails (N- and C-terminals) fluctuate more than any other part of the protein. Secondary structure elements such as alpha helices and beta strands are usually more rigid than the unstructured part of the protein and thus fluctuate less than the loop regions [16]. In the NS2B/NS3pro-ZINC000001680989, NS2B/NS3pro-ZINC000001684485 and NS2B/NS3pro-ZINC000006694490 complexes, the rate of fluctuation is higher at residue No. 32 compared to that in the NS2B/NS3pro-ZINC000001679427 complex. In the other residues, the fluctuations of the four complexes remained almost the same. All of the protein–ligand interactions had a fluctuation rate of less than 2 Å, which is completely acceptable. The complexes were considered stable throughout the MD simulation period. These plotted data indicate that all of the complexes showed active interaction with the protein and remained stable at the allosteric site during the simulation. It is interesting to note that, in the native NS2B/NS3 protein (black line), the RMSD fluctuation is significantly greater, showing the importance of the inhibitors effect at the system’s stabilization, which suggests NS2B/NS3 enzyme inactivation.

The calculation of the radius of gyration (Rg) was performed to evaluate the protein–ligand systems’ stability by calculating the structural compactness along the MD trajectories [24]. After the MD simulation calculation, the calculation of Rg was also used to determine the stably folded or unfolded protein and complexes system. The average values of RG for all of the complexes were calculated and were around 16.032, 16.111, 16.124 and 16,093 Å for NS2B/NS3Pro-ZINC000001679427, NS2B/NS3Pro-ZINC000001680989, NS2B/NS3Pro-ZINC0001684485 and NS2B/NS3Pro-ZINC00000694490, respectively (Figure 5). All of the complexes exhibited relatively similar and consistent values of Rg, indicating systems stability.

The solvent accessible surface area (SASA) was also determined to evaluate the constancy of simulated systems and to determine how much of the receptor area was exposed to the solvents. A larger SASA value indicates the growth of protein volume during the MD simulation. The interaction of the ligands in the active site can change the SASA and protein folding [25]. In the case of the complexes NS2B/NS3pro-ZINC000001679427 and NS2B/NS3pro-ZINC000001680989, it was found to be around 8556.41 A^2^ and 8645.09 A^2^, respectively (Figure 6). Likewise, the complexes NS2B/NS3pro-ZINC000001684485 and NS2B/NS3pro-ZINC000006694490 showed average values of SASA around 8916.33 A^2^ and 8711.76 A^2^. These calculations suggest that all of the complexes were least exposed to the water solvent during the 100 ns MD simulations, which indicates the relatively stable nature of these complexes.

To better understand the mode of interaction of these new inhibitors, it is important to estimate their binding affinities (Δ*G*_bind_) and the values of the other energy contributions. Hence, the MM-GBSA and MM-PBSA methods were used to calculate the free energy. These methods have been used successfully in the reproduction of experimental results [26] and in studies of the allosteric inhibitors of DENV NS2B/NS3pro [12].

The Δ*G*_bind_ and the values of the energy contributions of van der Waals (Δ*E*_vdW_), electrostatic (Δ*E*_ele_), polar (Δ*G*_GB_) and nonpolar (Δ*G*_nonpol_) forces are summarized in Table 2 (all results in Kcal/mol). Using the MMGBSA approach, the Δ*G*_bind_ contributions ordered the binding affinity of the ligands to the DENV NS2B/NS3pro allosteric site as ZINC000001680989 > ZINC000001679427 > ZINC000006694490 > ZINC000001684485. The MMPBSA approach ordered them as ZINC000001680989 > ZINC000006694490 > ZINC000001679427 > ZINC000001684485 (Table 2). This classification is in accordance with the values from the molecular docking, which suggests the same classification for these compounds.

The results obtained in this study demonstrate that the ligands coupled to the NS2B/NS3pro allosteric site had greater affinity compared to the results obtained using the free energy by Hariono et al. [18], where the matrix ligand presented an energy of −16.37 ± 3.22 kcal/mol by the MMPBSA method. This suggests the potential of these compounds to inhibit the allosteric site in study. Mukhametov et al. [12], in their in silico and in vitro studies of DENV NS2B/NS3pro allosteric inhibitors, predicted the Δ*G*_bind_ values of the compound 2-hydroxy3-methoxybenzaldehyde N-adenosinylhydrazine (compound 10) protease (65.1%) to be −24.02 kcal/mol and −14.45 kcal/mol, respectively, by the MMGBSA and MMPBSA methods. The Δ*G*_bind_ results for the ligands ZINC000001680989, ZINC000001679427 and ZINC000006694490 obtained in this study indicate the better affinity of these ligands with DENV NS2B/NS3pro (Table 2), a fact that refocuses our hypothesis regarding allosteric inhibitors. In general, the protein–ligand interaction is stabilized by several types of weak interactions, and the hydrogen bond interaction is potentially one of the most important. Table 3 shows the hydrogen bonds between the ligands ZINC000001680989, ZINC000001679427, ZINC000006694490 and ZINC000001684485, with residues from the NS2B/NS3pro allosteric site.

The Leu179(149), Ala 194(164) and Ans182(152) residues in NS2B/NS3pro played an important role in the formation of hydrogen bonds that remained occupied for more than 100%, 99% and 23% of the simulation time with ZINC000001680989, respectively. The ligand ZINC000001679427 also formed hydrogen bonds with the residues Ala194(164), Leu179(149) and Ans182(152) and was occupied for more than 73%, 50% and 14% of the simulation time, respectively. The ligand ZINC000006694490 interacted with the Asn182(152) and Lys104(74) residues and was occupied for more than 62% and 17% of the simulation time, respectively. For the ligand ZINC000001684485, the most significant interaction was with the residue Lys104(74), and it was occupied for 100% of the simulation time.

It is important to highlight that, in the study by Hariono et al. [18], the matrix compound presented the highest occupancy of hydrogen bonds during the simulation time, 60%, a lower number than the numbers the ligands ZINC000001680989, ZINC000001679427, ZINC000006694490 and ZINC000001684485 presented with the residues of the DENV NS2B/NS3pro allosteric site.

Therefore, the greater occupation of the hydrogen bonds of ZINC000001680989 with the NS2B/NS3pro allosteric residues may explain the better bond-free energy observed for the DENV NS2B/NS3pro-ZINC000001680989 complex (−40.65 ± 0.09 kcal/mol) compared to those observed for the DENV NS2B/NS3pro-ZINC000001679427 (−35.53 ± 0.11 kcal/mol), DENV NS2B/NS3pro-ZINC000006694490 (−27.76 ± 0.09 kcal/mol, and DENV NS2B/NS3pro- ZINC000001684485 (−11.7 ± 0.14 kcal/mol) complexes.

It is important to highlight that the interactions of ZINC000001680989 and ZINC000001679427 with the residues Leu179(149), Asn182(152) and Ala194(164) of the NS2B/NS3pro allosteric site had already been observed in the results of the molecular docking. Mukhametov et al. [12], in their in silico and in vitro studies on DENV NS2B/NS3pro allosteric inhibitors, reported that the most potent inhibitor (compound 10) formed hydrogen bonds with the residues Leu179(149), Asn182(152) and Ala194(164), which proved the high binding potential of the two ligands to the NS2B/NS3pro allosteric site. As was already mentioned, the hydrogen bond interaction with the Asn182(152) residue is considered essential in the inhibition process and corroborates the in silico and in vivo studies of [17,27]. The residues Leu149, Asn152 and Ala164 in this protease play key roles in the allosteric inhibition of the active site when binding with the ligand by bringing out significant conformational changes and disabling the protein to make its function [12].

In order to assess which residues contributed to the stabilization and maintenance of the simulated compounds in the NS2B/NS3pro allosteric site, an analysis of the free energy decomposition by the residue was conducted (Figure 7). In the graph, positive values correspond to unfavorable interactions, while negative values correspond to favorable interactions, which are those that contributed to the stabilization of the ligand in the complex.

An analysis of the interaction energy graph for every residue revealed that the residues that contributed most significantly to the total interaction energy and, therefore, to the stabilization of the complex were Lys73, Lys74, Leu76, Trp83, Leu85, Ile123, Val147, Gly148, Leu149, Asn152, Val154, Ala164, Ile165, Ala166 and Asn167 for ZINC000001680989. Likewise, Lys73, Lys74, Leu76, Trp83, Leu85, Ile123, Asn152, Val154, Ala164, Ile165 and Ala166 contributed most significantly for ZINC000001679427. For ZINC000006694490, the residues Lys74, Asp75, Leu76, Trp83, Ile123 and Asn152 were observed, as also evinced by the hydrogen bond (Table 3), Val154, Ala164 and Asn167. The ligand ZINC000001684485 showed an unfavorable interaction with the Glu88 residue and a favorable interaction with the residues Lys74, Leu76, Gly148, Leu149, Asn152, Ile165 and Ala166.

The per-residue interaction profile indicates that the four ligands present favorable interactions with the residues from the NS2B/NS3pro allosteric site, which is crucial for the protein–ligand complex’s stability. The results of the MD simulation and the MM/GBSA analysis validate the molecular docking results. The negative binding free energy of all four complexes demonstrates their stable configuration, which indicates that these compounds have enough affinity with NS2B/NS3pro DENV to be considered as allosteric inhibitors.

The greater number of residues with significant contributions from the ZINC000001680989 binder also corroborated the hypothesis of it having greater binding affinity. The hydrogen bond interactions, together with the hydrophobic clusters, contributed to the greater affinity of ZINC000001680989 bonding, with special emphasis on the hydrogen bond formed with the Leu179 residue, which also contributed to ZINC000001679427.

In particular, by analyzing the ligand ZINC000001680989, which showed the best binding affinities (Δ*G*_bind_) based on the MM/GBSA method, it was possible to observe interactions with the residues Lys73, Lys74, Leu149 and Asn152, which, according to experimental and theoretical studies, are essential for protease inhibition [12,13,17,27]. Othman and collaborators [28] observed that interactions with the residues Lys74 and Leu149 explained the differences in the inhibition activities of non-competitive inhibitors in their studies. This interaction with Lys74 is directly linked to Asp75, inducing a change in the region of the catalytic triad. This, presumably, could interrupt the electron transfer process necessary for the binding of the substrate at the active site, thus affecting the activity of the protease. Thus, the results obtained reinforce our strategy of using this class of compounds as allosteric inhibitors of DENV NS2B/NS3pro.

Another highlight was that the residues of Lys74, Leu76 and Asn152 appeared in all of the simulated systems. This indicated that the interaction of these residues with the ligands in the allosteric site was essential for their maintenance and stability. It is possible to explore these interactions in future studies.

It is interesting to note that, when we have conducted visualization of the frames corresponding to the times of 0, 25, 50, 75 and 100 ns, we have noted that the ligands ZINC000001680989, ZINC000006694490 and ZINC000001679427, according to the MMGBSA energies, remain in the enzyme active site (Figure 8).

### 2.3. SwissADME Prediction of Selected Compounds

The data predicted for the lipophilicity, druglikeness, solubility, physicochemical characteristics, pharmacokinetics and medicinal chemistry of the selected compounds evaluated by SwissADME are given in Table 4. According to Lipinski′s rule of five, the molecular weights of the selected compounds reached between 313.42 Da (ZINC000001679427) and 369.44 Da (ZINC000001684485), respectively, and within the limits up to 500 Da. The log Po/w values were less than 5 and were in the range of 2.33 (ZINC000001684485) up to 3.50 (ZINC000001680989), suggesting a good potential for lipophilicity. The HBA number of all the selected products was between 3 and 5 and less than 10; the number of HBD atoms was 3 (ZINC000006694490) and 1 for all other hits, respectively—both less than 5 [29,30].

For physicochemical properties such as molecular refractivity, the hit ZINC000001680989 showed a value of 101.23, indicating a greater size and polarizability. The surface sum over all of the polar atoms or molecules, primarily oxygen and nitrogen, including their attached hydrogen atoms, was evaluated from the topological polar surface area (TPSA) [31], in which the hit ZINC000006694490 exhibited an index of 120.89, the highest among the compounds selected.

The potential of the selected hits for water solubility was evaluated from Log S (SILICOS-IT), a hybrid method relying on 27 fragments and 7 topological descriptors. The solubility class: Log S Scale: insoluble < −10 poorly < −6, moderately <−4 soluble < −2. Only the hit ZINC000001680989 showed a low potential for solubility in water, with a Log S = −6.02. All of the other compounds exhibited potential values for moderate solubility [32].

For the druglikeness properties, the evaluated hits did not show any variation from Linpinski′s rules, and all of the hits had a bioavailability score of 0.55. The bioavailability score is similar but seeks to predict the probability of a hit having at least 10% oral bioavailability in rats or measurable Caco-2 permeability. This semi-quantitative rule-based score relying on total charge, TPSA and the violation of the Lipinski filter defines four classes of compounds with probabilities of 11%, 17%, 56% and 85% [30,33].

The skin permeation coefficient (kp) in pharmacokinetics calculates the conductance of skin to a particular chemical from a specific aqueous vehicle using three different models. The ZINC000001684485 presented the lowest value (−6.91) in the series of compounds, while the index of −5.17 (ZINC000001679427) can be considered the highest in the series. It is worth noting the little statistical significance between the values of all the selected hits.

All of the selected compounds showed negative indices for the potential penetration of the blood brain barrier. On the other hand, all of the hits studied expressed high human intestinal absorption (GI). All of the compounds can be promising agents that can very easily be absorbed by the gastrointestinal tract without potential BBB permeability. Since these compounds cannot cross the BBB, they do not cause central nervous system depression or drowsiness as side effects [30].

Regarding the parameters of SwissADME medicinal chemistry, the synthetic accessibility (SA) score is basically based on the assumption that the frequency of molecular fragments in ‘really’ obtainable compounds correlates with the ease of synthesis. The fragmental contribution to SA should be favorable for frequent chemical moieties and unfavorable for infrequent moieties. The scale score is normalized to range from 1 (very easy) to 10 (very difficult to synthesize) [34]. All of the compounds selected in this study showed indexes lower than 4, especially ZINC000001679427, which had an index lower than 3, corroborating the presence of chemical groups that are not rare in the structure of the selected molecules and suggesting the ease of new syntheses.

### 2.4. Toxicological Analysis by Derek Nexus^TM^

The compound ZINC000006694490 exhibited a warning with a plausible level of toxicity for the purine base chemical moiety. In this analysis, reports were identified in the literature for hepatotoxicity and chromosome damage in vivo in mouse, rat and human organisms.

This alert describes the hepatotoxicity of purine derivatives. Purines have been shown to cause severe liver toxicity in humans and animals. Steatosis, cholestatic jaundice, necrosis and veno-occlusive disease were reported in patients treated with purine derivatives during immunosuppression or antiretroviral therapies [35]. In animals, elevations of the liver enzymes and damage of the centrilobular hepatocytes were observed [36]. The mechanism of toxicity of purines has dose-dependent and accumulative characteristics, but hypersensitivity reactions have also been reported after a short time of treatment [37].

The prolonged exposure of HIV-positive patients to didanosine (daily dose: 400 mg) during antiretroviral therapy led to the elevation of liver enzymes, various degrees of micro or macrovesicular steatosis and mild cholestasis [38,39]. Centrilobular Mallory bodies have also been observed [40]. In children, therapy with didanosine (120–270 mg/m^2^) caused severe hepatic injury such as necrotising hepatitis, which in some cases led to fatal hepatic failure [41]. Patients undergoing immunosuppression therapy with azathioprine (daily dose: 50–2500 mg) for the management of autoimmune diseases such as inflammatory bowel disease, rheumatoid arthritis, lupus erythematous and chronic hepatitis developed liver injury manifested with the elevation of liver enzymes, icterus and, in some cases, fever, diarrhea and vomiting [42,43,44]. The liver biopsies of these patients showed various degrees of liver damage from canalicular cholestasis, with ballooning centrilobular hepatocytes, necrosis with portal infiltrates and cirrhosis [43,44].

### 2.5. Chromosome Damage: In Vivo Chromosome Aberration Test; In Vivo Micronucleus Test

This alert describes the activity of exogenous purine base, nucleoside and nucleotide analogues in the in vivo chromosome aberration and micronucleus tests. Such compounds are widely used in medicine as antiviral or antineoplastic therapies and have been shown to induce chromosome damage in vivo. For example, positive results in the mouse bone marrow micronucleus test have been reported for acyclovir, ganciclovir and penciclovir when administered intravenously [45] and for 6-mercaptopurine [46] and 6-thioguanine [NTP] following the administration by oral gavage.

In addition, 6-mercaptopurine demonstrated activity when tested in the mouse bone marrow chromosome aberration test [47]. Didanosine (2′,3′-dideoxyinosine) failed to induce micronuclei in mouse bone marrow when administered orally but demonstrated activity following the administration by intraperitoneal injection [48]. This may be explained by the decomposition of didanosine under the acidic conditions of the stomach [49].

There are several mechanisms that may contribute to the clastogenicity of these compounds. Exogenous base and nucleoside analogues may become incorporated into mammalian DNA [50]. Once incorporated, the type of mechanism that leads to cytogenetic damage may be dependent upon the structural modification in the analogue. For example, ring-modified purines may pair with an incorrect nucleotide on the complementary DNA strand and induce mutations. By contrast, compounds with modified sugar moieties may distort DNA, causing repair and replication errors, or induce the premature termination of growing DNA chains [50].

## 3. Materials and Methods

### 3.1. Virtual Screening for Ligand Selection

Initially, we selected the S-[2-(pentanoylamino)-9H-purin-6-yl]pentanethioate(1), a thioguanine analogue (Figure 9) that was designed as a DENV NS2B/NS3pro inhibitor (IC50 of 0.38 μM) in a study conducted by Hariono and collaborators [18]. This compound served as a matrix in the present study.

After that, compounds structurally similar to 1 were searched for using the Swiss Similarity web tool (http://ww.swisssimilarity.ch/, accessed on 10 April 2022) [51] and using the FP2 fingerprints method to screen the library of lead-like molecules from ZINC in order to suggest compounds with an inhibition potential of DENV NS2B/NS3pro by means of allosteric modulation, since the binding of molecules to the allosteric site causes conformational changes, reduces motion reflected in fluctuation and indicates the possibility of terminating protein function. This search selected 71 compounds.

Swiss Similarity is based on the LBVS hypothesis, and similar molecules are prone to exhibit similar biological activity. This tool allows for the rapid screening of large-scale drug libraries in more than 30 chemical databases including more than 205 million virtual compounds synthesizable from commercially obtained reagents [51]. This tool has been used for in silico studies in drug planning [52].

### 3.2. Molecular Docking

The 71 ligands selected in the previous step were subjected to molecular coupling. The DENV NS2B/NS3pro crystallographic model was retrieved from the Protein Data Bank with the code PDB 2FOM and a resolution of 1.50 Å [19]. To perform the molecular docking simulations, the CSDGOLD program [53] was used, and the docking grids with a 10 Å radius were positioned on the specific allosteric binding site of the non-competitive inhibitor DENV NS2BNS3pro according to [12,13]. For the CSD-GOLD protocol, we set the following parameters for the ChemPLP algorithm: all water molecules and ions have been removed. The CSDGOLD program uses the empirical aptitude function called ChemPLP, which consists of the application of terms of hydrogen and metal bonding and linear potential by parts (PLP) to model the steric complementarity between the protein and the ligand. The dimensionless scoring scale measures the success of the pose; higher scores indicate better nesting positions [54].

### 3.3. Molecular Dynamics Simulation

The best docking results will be subjected to molecular dynamics (MD) simulation in order to monitor the behavior of the systems in the function of time [55]. The impact of MD simulations on molecular biology and drug discovery has expanded dramatically in recent years. These simulations capture the behavior of proteins and other biomolecules completely at the atomic level and in very fine temporal resolutions. Major improvements in terms of simulation speed, accuracy and accessibility, along with the proliferation of structural and experimental data, have increased the search for biomolecular simulations for experimentalists [56].

To analyze the conformational changes of the protein and ligand structures, as well as the stability of the ligand-receptor complexes, we performed MD simulations, accelerating the graphics processing unit (GPU) in Amber18 [57]. The Amber ff14SB [58] and the general amber force field (GAFF) [59] were selected as the parameters to describe the protein and ligands, respectively. The restrained electrostatic potential (RESP) charges for the ligands were calculated at the HF/6-31G(d) level of theory using the Gaussian 09 program [60]. The systems were solvated in a cubic water-box with the explicit solvation model TIP3P using a distance of 12.0 Å with periodic boundary conditions. The Na+ ions were added to maintain the electroneutrality of the systems. To eliminate the steric clashes, each system was subjected to three minimization steps. First, all hydrogen atoms were minimized by 2000 steps of the steepest descent, followed by 3000 steps of the conjugate gradient algorithm. Next, using the same protocol, the positions of the water molecules were relaxed. Lastly, the whole system was energy-minimized for 5000 steps of the steepest descent plus 5000 steps of the conjugate gradients. Afterward, we started the heat of the system from 0 to 300 K, running 200 ps of MD and then 300 ps of density balance with position restraints on the protein–ligand atoms at a constant volume. Before performing the production step, all of the protein–ligand systems were balanced with 500 ps of MD without positional restraints at a constant pressure. The temperature was maintained at 300 K by coupling to a Langevin thermostat using a collision frequency of 2ps-1. A cutoff of 10 Å was employed for the nonbonded interactions, and the Particle Mesh Ewald (PME) [61] method and the SHAKE [62] algorithm were used to restrict the bond lengths involving the hydrogen atoms. The MD simulations (production) were performed using 100 ns at a temperature of 300 K. The structural analysis of each system was performed by using the root means square deviation (RMSD) values of the heavy atoms of the protein′s backbones, the root mean square fluctuation (RMSF), the radius of gyration (RG) and the solvent accessible surface area (SASA) obtained by the CPPTRAJ module of Amber18.

### 3.4. Binding Free Energy Calculations MM/PBSA and MM/GBSA

The values of free binding energy (Δ*G**_bind_*) for each system were calculated using the MM/PBSA and MM/GBSA approaches implemented in the MMPBSA.py module. For this, the last 10 ns of the DM simulation were used.

The equations that describe the calculations of the energy are:∆*G_bind_* = ∆*H* − *T*∆ ≈ ∆*E_MM_* + ∆*G_sol_* + T∆*S*,(1)
∆*E_MM_* = ∆*E_internal_* + *E*∆*_electrostatic_* + ∆*E_vdw_*
(2)
∆*G_sol_* = ∆*G_PB/GB_* + ∆*G_SASA_*
(3)

As shown in Equation (1), the enthalpy part is expressed as the summation of the molecular mechanical energy (∆*E_MM_*) and the solvation free energy (∆*G**_sol_*), where ∆*E_MM_* is composed of the intra-molecular energy (∆*E**_internal_*, including the bond, angle and dihedral energies of the system), the electrostatic energy (∆*E_ele_*) and the van der Waals interactions (∆*E_vdW_*) (Equation (2)). The solvation free energy (∆*G**_sol_*) is also composed of two parts, namely, the polar part (∆*G**_pol_*) and the non-polar (∆*G_np_*) part (Equation (3)), where ∆*G_pol_* is usually computed by the Generalized Born (GB) model or by solving the Poisson–Boltzmann (PB) equation, while ∆*G_np_* is estimated by the solvent accessible surface area (SASA)-based approach [26,63].

### 3.5. Decomposition Energy by Residue

The method of the decomposition of energy by residue was used to determine the total energy contribution of each residue to the drug–receptor interaction and also to investigate the chemical nature of its interactions [64].

The interaction energy of the residues can be described from four terms: the electrostatic contribution of van der Waals (Δ*E_vdw_*), the electrostatic contribution (Δ*E_ele_*) in the gas phase, the contribution of polar solvation (Δ*G_pol_*) and the contribution of non-polar solvation (Δ*G_nonpol_*), according to the equation:∆*G_inhibitor_* = ∆*E_vdw_* +∆*E_ele_* + ∆*G_pol_* + ∆*G_nonpol_*(4)

### 3.6. Pharmacokinetics Analysis by SwissADME

The computational studies of the compounds from the ZINC database 1–3 were selected to predict the molecular properties using the SwissADME online server (http://www.swissadme.ch/, accessed on 10 April 2022) [65]. The molecular properties such as the volume, molecular weight, logarithm of partition coefficient (Log Po/w), number of hydrogen-bond donors (HBDs), number of hydrogen-bond acceptors (HBAs), molar refractivity (MR), topological polar surface area (TPSA), water solubility (Log S), skin permeation (Log Kp), bioavailability score (BS), synthetic accessibility (SA), number of rotatable bonds and Lipinski’s rule of five of the selected compounds were evaluated [66,67].

### 3.7. Toxicity Analysis by DerekTM

The predictions described below were made on a computer with six logical processors of an Intel^®^ Core TM i5-9400, 4.10 GHz processor using the Windows 10 Professional operating system.

Derek Nexus is an ‘expert rule-based’ system for the prediction of toxicity [68,69,70] that uses a virtual base of molecules with reported toxicity alerts—for example, mutagenicity, genotoxicity, teratogenicity, carcinogenicity and hepatotoxicity—for comparison with the potential toxic group present in each query molecule [6]. The program generates an alert based on the literature evidence and describes the potential toxicity for the complete structure [33].

## 4. Conclusions

In the present study, in order to investigate the binding affinity, selectivity and stability of candidates for allosteric inhibitors of the DENV NS2B-NS3pro enzyme, the Swiss Similarity web tool was utilized in order to search for compounds structurally similar to ligand 1 described by Hariono et al. Within this web server, 71 compounds were selected for molecular docking studies using the GOLD program. The four best-scored ligands according to the scoring function energies were then subjected to an MD protocol involving 100 ns of simulation in the AMBER package. The free bond energies, hydrogen bonds and energy contributions per residue were also calculated for the simulated systems. The computational approach used here proved to be useful for planning new inhibitors to combat dengue. In the simulations, the residues of Lys74, Leu176 and Asn152 contributed to the stability of the ligand interactions at the allosteric site in all of the simulated systems. Meanwhile, the Leu149 and Ala166 residues, in addition to contributing significantly to the energy decomposition by the residue, showed high rates of hydrogen bond permanence. This indicated that these interactions are essential for maintenance and stability.

The selected compounds showed negative indices for the potential penetration of the blood brain barrier and expressed high human intestinal absorption. They can be promising agents that can very easily be absorbed by the gastrointestinal tract without potential BBB permeability, indicating a low risk of central nervous system depression or drowsiness as side effects. All of the compounds selected in this study showed synthetic accessibility (SA) scores lower than 4, especially ZINC000001679427, with an index lower than 3, corroborating the presence of chemical groups that are not rare in the structure of the selected molecules and suggesting the ease of new syntheses.

In the toxicological analysis, the ZINC000006694490 compound exhibited an alert with a plausible level of toxicity for the purine base chemical moiety, indicating hepatotoxicity and chromosome damage in vivo in mouse, rat and human organisms. These results agree with those of previous studies. Hence, there is scope for further work related to the selective planning of candidates for DENV NS2B-NS3pro inhibitors.

## Figures and Tables

**Figure 1 molecules-27-04118-f001:**
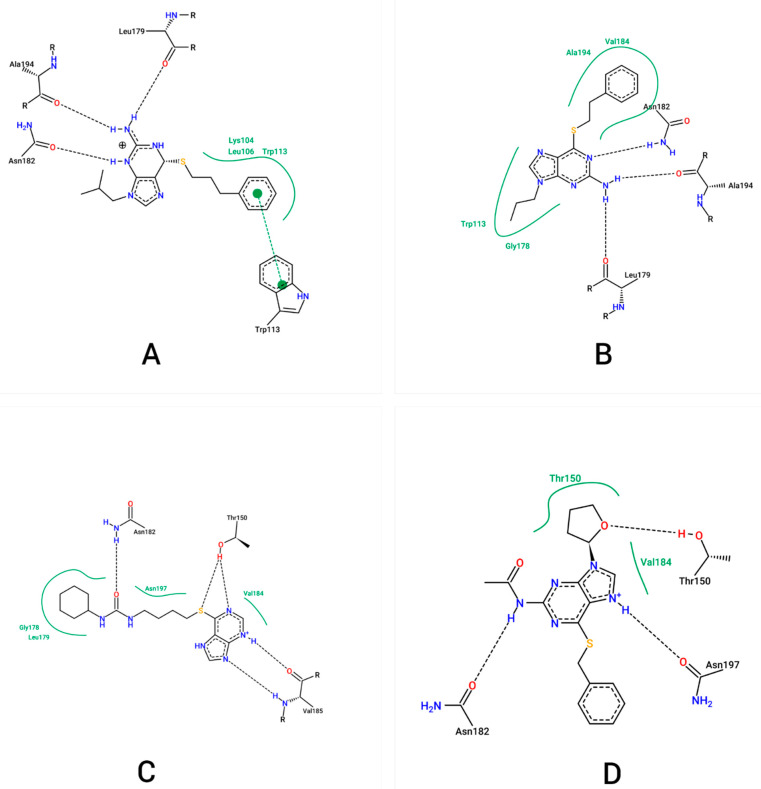
Predicted interactions in the docking with the NS2B/NS3pro enzyme of the ligands ZINC000001680989 (**A**), ZINC000001679427 (**B**), ZINC000006694490 (**C**) and ZINC000001684485 (**D**) in the GOLD program. The dashed black lines represent the hydrogen bonds and the green lines represent the hydrophobic interactions. The images were obtained through the PoseView Web 1.97 server.

**Figure 2 molecules-27-04118-f002:**
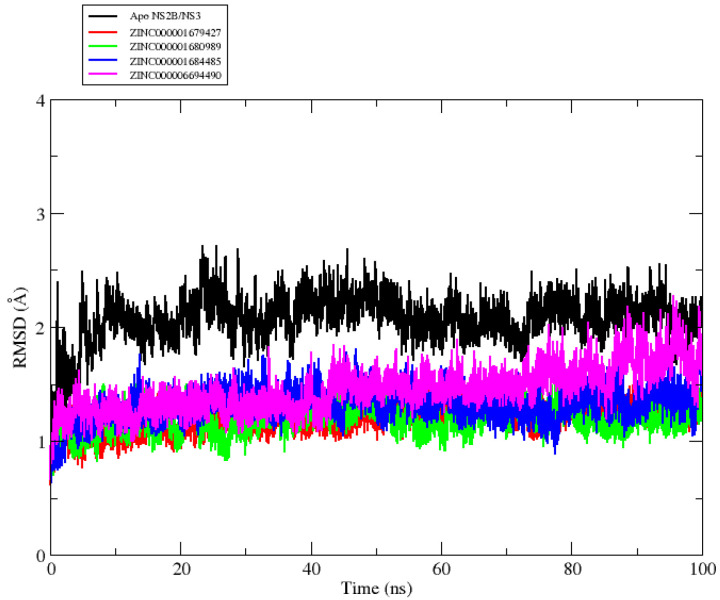
. Graphical representation of the RMSD values versus the simulation time to the backbone Apo NS2B/NS3 (black line) ZINC000001679427 (red line), ZINC000001680989 (green line), ZINC000001684485 (blue line) and ZINC000006694490 (magenta line) studied.

**Figure 3 molecules-27-04118-f003:**
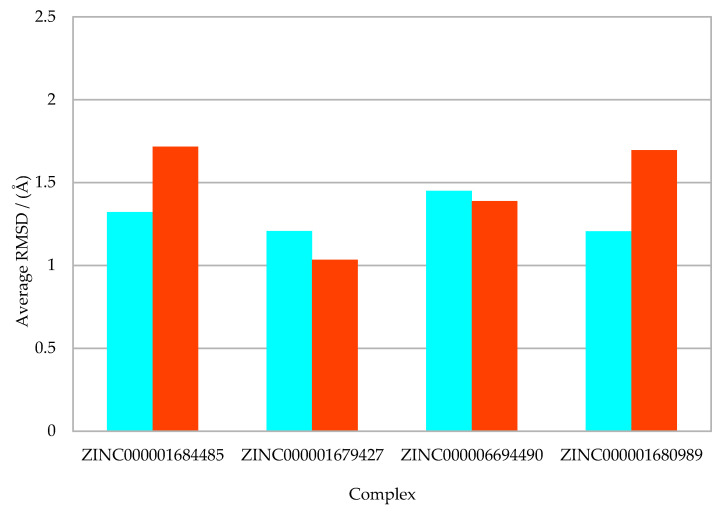
Average RMSD values between protein/protein (orange) and ligand/ligand (cyan).

**Figure 4 molecules-27-04118-f004:**
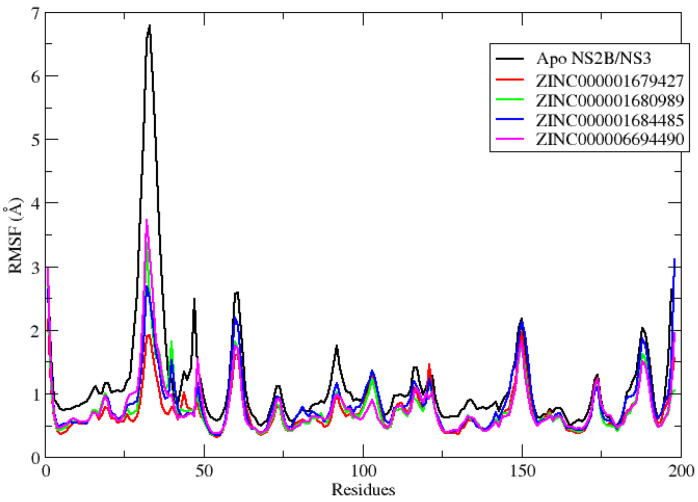
Graphical representation of the RMSF values versus the simulation time of the NS2B/NS3pro Apo backbone (black line) and its complex with the ZINC000001679427 (red line), ZINC000001680989 (green line), ZINC000001684485 (blue line) and ZINC000006694490 (magenta line) complexes studied.

**Figure 5 molecules-27-04118-f005:**
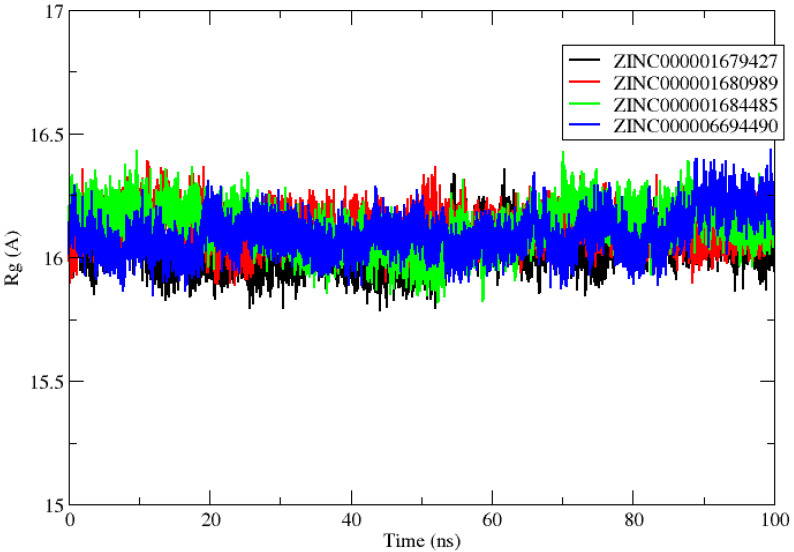
Radius of gyration (Rg) of the NS2B/NS3pro ZINC000001679427 (black line), ZINC000001680989 (red line), ZINC000001684485 (green line) and ZINC000006694490 (blue line) complexes studied.

**Figure 6 molecules-27-04118-f006:**
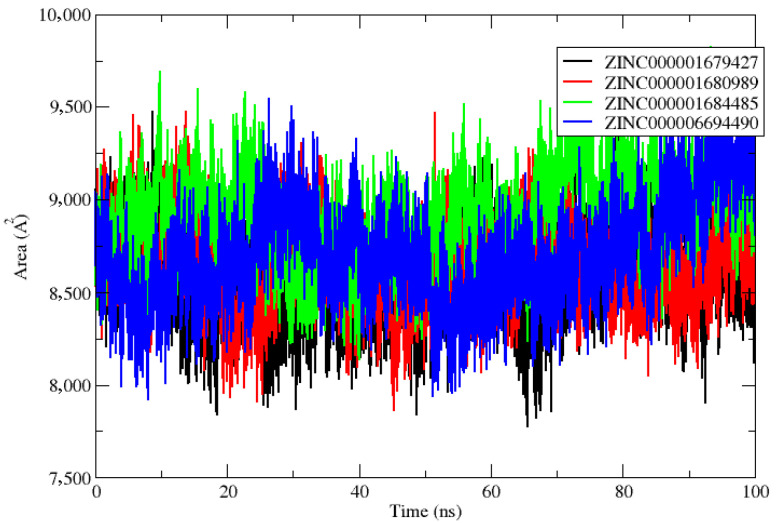
Solvent surface accessible area (SASA) of the NS2B/NS3pro ZINC000001679427 (black line), ZINC000001680989 (red line), ZINC000001684485 (green line) and ZINC000006694490 (blue line) complexes studied.

**Figure 7 molecules-27-04118-f007:**
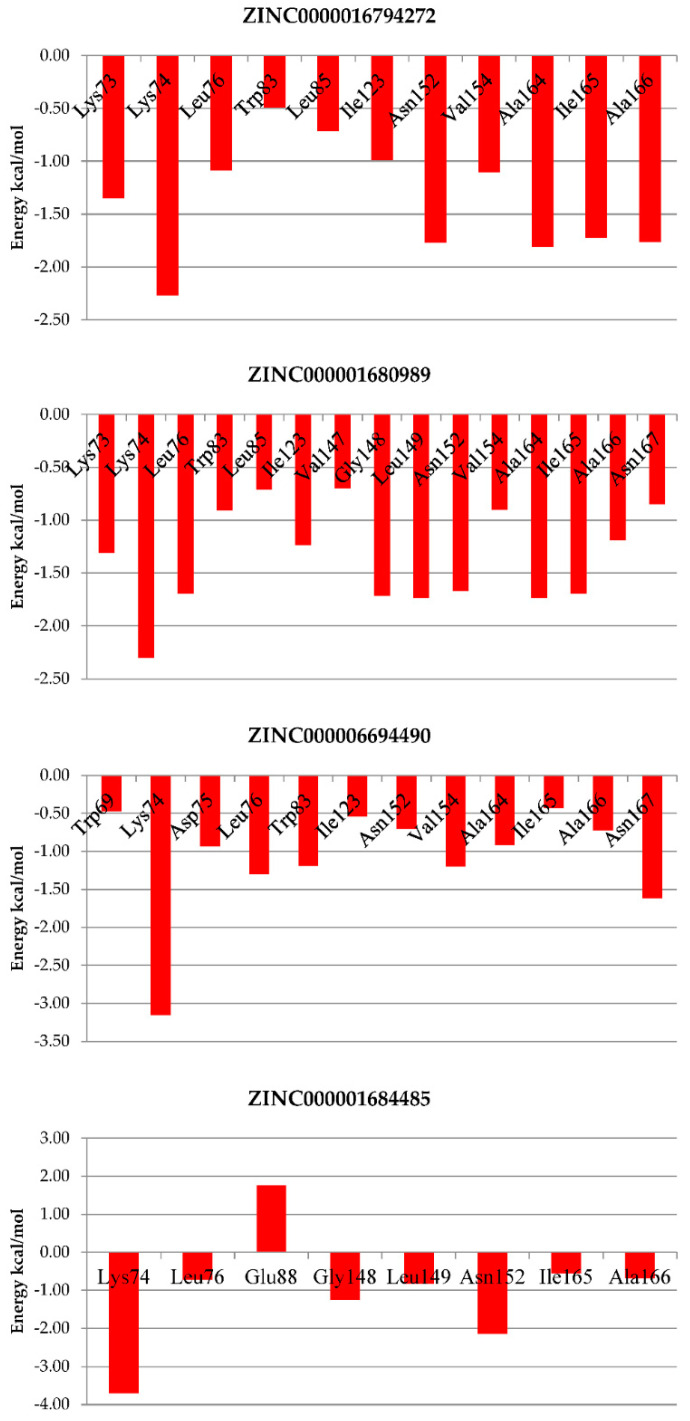
Graphical representation of the interaction energy per residue.

**Figure 8 molecules-27-04118-f008:**
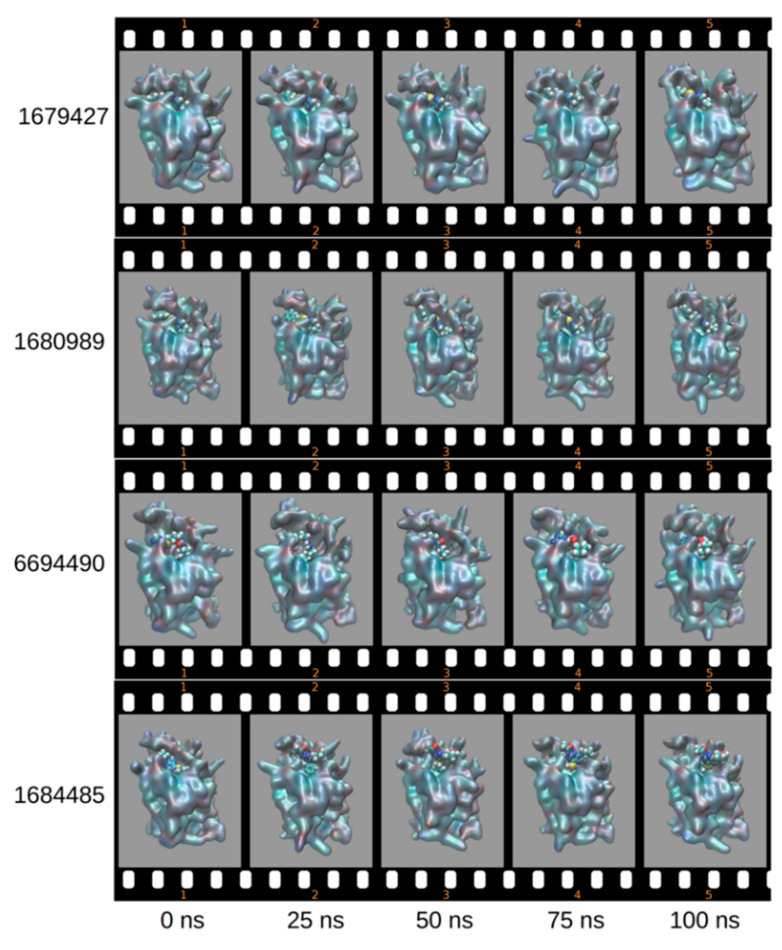
Frames in the function of time of the enzyme/ligand complexes.

**Figure 9 molecules-27-04118-f009:**
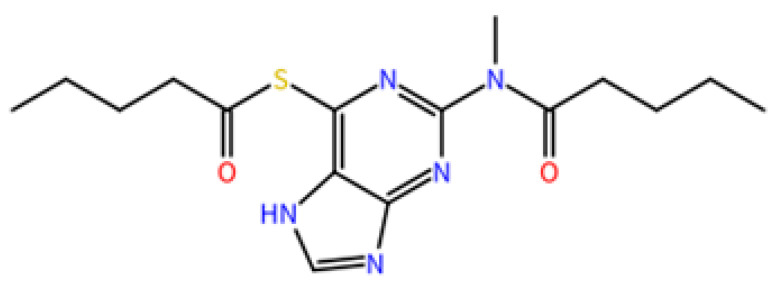
Structure of matrix compound 1.

**Table 1 molecules-27-04118-t001:** PLPChem score values obtained in the GOLD program and the structures of the four selected compounds.

Ligands	PLPchem	Structures
ZINC000001680989	62.05	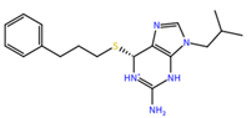
ZINC000001679427	59.69	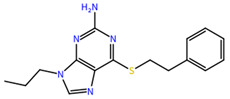
ZINC000006694490	58.41	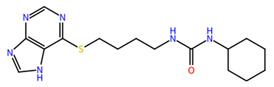
ZINC000001684485	58.36	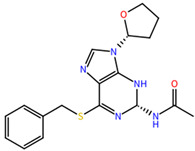

**Table 2 molecules-27-04118-t002:** Estimated binding affinities and energetic contributions for the systems under study.

Ligand ID	Terms					
	ΔE_vdW_	ΔE_ele_	ΔG_GB_	ΔG_nonpol_	ΔG_MMGBSA_	ΔG_MMPBSA_
ZINC000001680989	−44.83 ± 0.10	−21.84 ± 0.11	31.28 ± 0.09	−5.15 ± 0.01	−40.65 ± 0.09	−39.43 ± 0.12
ZINC000001679427	−42.58 ± 0.10	−21.53 ± 0.19	33.98 ± 0.16	−5.40 ± 0.01	−35.53 ± 0.11	−25.72 ± 0.13
ZINC000006694490	−38.22 ± 0.09	−33.08 ± 0.25	48.50 ± 0.20	−4.96 ± 0.01	−27.76 ± 0.09	−30.30 ± 0.12
ZINC000001684485	−22.14 ± 0.07	−31 11± 0.15	45.84 ± 0.35	−4.31 ± 0.01	−11.71 ± 0.14	−19.67 ± 0.22

**Table 3 molecules-27-04118-t003:** Hydrogen bonds between the compounds ZINC000001680989, ZINC000001679427, ZINC000006694490 and ZINC000001684485 with NS2B/NS3pro, which found at least 2.0% occupancy throughout 100 ns of simulation time.

Complex	Hydrogen Bond Formation	Distance (Å)	Occupancy (%)
NS2B/NS3pro-ZINC000001680989	LEU179@O-LIG@H16-LIG@N3	2.82	86.39
	ALA194@O-LIG@H15-LIG@N3	2.83	82.74
	LIG@N2-ASN182@HD21-ASN_182@ND2	2.91	20.09
	ALA194@O-LIG@H16-LIG@N3	2.82	16.96
	LEU179@O-LIG@H15-LIG@N3	2.82	16.85
	LIG@N4-LEU179@H-LEU179@N	2.94	8.91
	LIG@N3-ASN182@HD21-ASN182@ND2	2.93	2.49
NS2B/NS3pro-ZINC000001679427	ALA194@O-LIG@H15-LIG@N3	2.83	46.96
	LEU179@O-LIG@H14-LIG@N3	2.83	32.68
	ALA194@O-LIG@H14-LIG@N3	2.83	26.39
	LEU179@O-LIG@H15-LIG@N3	2.82	15.65
	LIG@N3-ASN182@HD21-ASN182@ND2	2.92	11.13
	LIG@N4-ASN182@HD21- ASN182@ND2	2.93	3.37
	LIG@N2-LEU179@H-LEU179@N	2.94	2.58
	LYS104@O-LIG@H14-LIG@N3	2.82	2.37
NS2B/NS3pro-ZINC000006694490	ASN182@OD1- LIG@H23-LIG@N4	2.82	62.53
	LYS104@O-LIG@H23-LIG@N4	2.83	8.58
	ILE195@O-LIG@H11-LIG@N5	2.88	6.83
	ILE_195@O-LIG@H-LIG@N	2.88	5.58
	LIG@N1-ASN197@HD21-ASN197@ND2	2.91	4.47
	LIG@O-LYS104@HZ1-LYS104@NZ	2.82	3.27
	LIG@OLYS104@HZ2-LYS104@NZ	2.81	3.19
	LIG@O-LYS104@HZ3-LYS104@NZ	2.81	2.86
NS2B/NS3pro-ZINC000001684485	LYS104@NZ-LIG@H-LIG@N	2.80	95.48
	LIG@N-LYS104@HZ1-LYS104@NZ	2.80	55.70
	LIG@N-LYS104@HZ3-LYS104@NZ	2.80	20.84
	LIG@N-LYS104@HZ2-LYS104@NZ	2.79	18.93
	LIG@N4-ASN197@HD21-ASN197@ND2	2.91	2.04

**Table 4 molecules-27-04118-t004:** Lipophilicity, druglikeness, solubility, physicochemical characteristics, pharmacokinetics and medicinal chemistry of the selected compounds evaluated by SwissADME.

Ligands	MR	TPSA (Å2)	Log Po/w	Log S	Log Kp (cm/s)	BS	SA
ZINC000001684485	100.5	107.23	2.33	Moderately soluble	−6.91	0.55	3.69
ZINC000006694490	95.55	120.89	2.40	Moderately soluble	−6.27	0.55	3.19
ZINC000001680989	101.23	94.92	3.50	Poorly soluble	−5.18	0.55	3.06
ZINC000001679427	91.62	94.92	2.87	Moderately soluble	−5.17	0.55	2.86

## Data Availability

Not applicable.

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
