# Peer review of "A Computational Approach Applied to the Study of Potential Allosteric Inhibitors Protease NS2B/NS3 from Dengue Virus"

_molecules, 2022, doi:10.3390/molecules27134118_

Round 1
Reviewer 1 Report
The manuscript entitled " A computational approach applied to the study of potential allosteric inhibitors protease NS2B/NS3 from Dengue Virus" aims to screen and identify Dengue virus NS2B/NS3 protease small molecule inhibitors using several computational tools. The manuscript concept is very similar to previously published studies for Dengue Virus NS2B/NS3 protease (Wichapong et al., 2010, J. Mol. Recognit.; Tahir ul Qamar et al., 2019, Sci. Rep.; Uday et al., 2021, Plos One; Mukhametov et al., 2014, J. Mol. Graph. Model.; etc.). The study identifies a few compounds that are expected to be promising and will benefit from experimental characterization, in vitro protease inhibition, and antiviral assays as well as cytotoxicity assay. Overall, the manuscript is well written but lacks experimental validation. The English grammatical revision will benefit the text.
I recommend authors to work on the following comments:
Authors should experimentally validate the identified small molecules using in vitro protease inhibition and antiviral assays.
In “Section 2.1 Molecular Docking”, lines 115-118, the authors mentioned that Asn182(152) residue is considered essential for interaction with NS2B/NS3pro allosteric inhibitors and is in accordance with other molecular coupling studies that involve DENV NS2B/NS3pro allosteric inhibitors. The authors should cite relevant references to support their statement.
In Molecular dynamics simulation, why authors haven’t analyzed the NS2B-NS3pro interaction in the absence and presence of predicted inhibitors? Authors should provide an insight on NS2B-NS3pro interaction in the manuscript.
Authors should investigate the stability of identified small molecules in the cellular environment.
Minor concerns
Line 74, correct it to “peptidomimetics”.
Line 118, put a full stop before “This suggests…”.
Lines 233-234, translate the sentence to English, and rewrite.
Line 545, Please cite the following reference for the PDB: 2FOM.
“Erbel P, Schiering N, D'Arcy A, Renatus M, Kroemer M, Lim SP, Yin Z, Keller TH, Vasudevan SG, Hommel U. Structural basis for the activation of flaviviral NS3 proteases from dengue and West Nile virus. Nat Struct Mol Biol. 2006 Apr;13(4):372-3. doi: 10.1038/nsmb1073. Epub 2006 Mar 12. PMID: 16532006.”
Throughout the manuscript, the authors refer to the "Swiss Similarity tool" as the Swiss Similary tool. Please make corrections.
Throughout the manuscript, the authors have used the terms “affinity energy” or “docking energy” but haven’t used the relevant unit of measurement (e.g., kcal/mol) for docking output. Authors should simply use the term “PLPChem score” instead of affinity or docking energy.
Author Response
Point 01
Comments and Suggestions for Authors
The manuscript entitled " A computational approach applied to the study of potential allosteric inhibitors protease NS2B/NS3 from Dengue Virus" aims to screen and identify Dengue virus NS2B/NS3 protease small molecule inhibitors using several computational tools. The manuscript concept is very similar to previously published studies for Dengue Virus NS2B/NS3 protease (Wichapong et al., 2010, J. Mol. Recognit.; Tahir ul Qamar et al., 2019, Sci. Rep.; Uday et al., 2021, Plos One; Mukhametov et al., 2014, J. Mol. Graph. Model.; etc.). The study identifies a few compounds that are expected to be promising and will benefit from experimental characterization, in vitro protease inhibition, and antiviral assays as well as cytotoxicity assay. Overall, the manuscript is well written but lacks experimental validation. The English grammatical revision will benefit the text.
I recommend authors to work on the following comments:
Authors should experimentally validate the identified small molecules using in vitro protease inhibition and antiviral assays.
In “Section 2.1 Molecular Docking”, lines 115-118, the authors mentioned that Asn182(152) residue is considered essential for interaction with NS2B/NS3pro allosteric inhibitors and is in accordance with other molecular coupling studies that involve DENV NS2B/NS3pro allosteric inhibitors. The authors should cite relevant references to support their statement.
In Molecular dynamics simulation, why authors haven’t analyzed the NS2B-NS3pro interaction in the absence and presence of predicted inhibitors? Authors should provide an insight on NS2B-NS3pro interaction in the manuscript.
Authors should investigate the stability of identified small molecules in the cellular environment.
Minor concerns
Line 74, correct it to “peptidomimetics”.
Line 118, put a full stop before “This suggests…”.
Lines 233-234, translate the sentence to English, and rewrite.
Line 545, Please cite the following reference for the PDB: 2FOM.
“Erbel P, Schiering N, D'Arcy A, Renatus M, Kroemer M, Lim SP, Yin Z, Keller TH, Vasudevan SG, Hommel U. Structural basis for the activation of flaviviral NS3 proteases from dengue and West Nile virus. Nat Struct Mol Biol. 2006 Apr;13(4):372-3. doi: 10.1038/nsmb1073. Epub 2006 Mar 12. PMID: 16532006.”
Throughout the manuscript, the authors refer to the "Swiss Similarity tool" as the Swiss Similary tool. Please make corrections.
Throughout the manuscript, the authors have used the terms “affinity energy” or “docking energy” but haven’t used the relevant unit of measurement (e.g., kcal/mol) for docking output. Authors should simply use the term “PLPChem score” instead of affinity or docking energy.
Response 01
Overall, the manuscript is well written but lacks experimental validation. The English grammatical revision will benefit the text.
Our research group develops theoretical works and, in addition, we have few resources for research in Brazil, which makes the feasibility of experimental validation impossible at the moment.
Authors should experimentally validate the identified small molecules using in vitro protease inhibition and antiviral assays.
Our research group develops theoretical works and, in addition, we have few resources for research in Brazil, which makes the feasibility of experimental validation impossible at the moment.
In “Section 2.1 Molecular Docking”, lines 115-118, the authors mentioned that Asn182(152) residue is considered essential for interaction with NS2B/NS3pro allosteric inhibitors and is in accordance with other molecular coupling studies that involve DENV NS2B/NS3pro allosteric inhibitors. The authors should cite relevant references to support their statement.
The following references were added:
https://doi.org/10.2298/JSC210929011D
https://doi.org/10.1016/j.jmgm.2014.06.008
https://doi.org/10.1021/acs.jmedchem.9b01697
In Molecular dynamics simulation, why authors haven’t analyzed the NS2B-NS3pro interaction in the absence and presence of predicted inhibitors? Authors should provide an insight on NS2B-NS3pro interaction in the manuscript.
Data added and discussed in the text.
Authors should investigate the stability of identified small molecules in the cellular environment.
Unfortunately we are not able to carry out this investigation.
Minor concerns
Line 74, correct it to “peptidomimetics”.
- Change made.
Line 118, put a full stop before “This suggests…”.
- Change made.
Lines 233-234, translate the sentence to English, and rewrite.
- Change made.
Line 545, Please cite the following reference for the PDB: 2FOM.
“Erbel P, Schiering N, D'Arcy A, Renatus M, Kroemer M, Lim SP, Yin Z, Keller TH, Vasudevan SG, Hommel U. Structural basis for the activation of flaviviral NS3 proteases from dengue and West Nile virus. Nat Struct Mol Biol. 2006 Apr;13(4):372-3. doi: 10.1038/nsmb1073. Epub 2006 Mar 12. PMID: 16532006.”
- Change made.
Throughout the manuscript, the authors refer to the "Swiss Similarity tool" as the Swiss Similary tool. Please make corrections.
- Change made.
Throughout the manuscript, the authors have used the terms “affinity energy” or “docking energy” but haven’t used the relevant unit of measurement (e.g., kcal/mol) for docking output. Authors should simply use the term “PLPChem score” instead of affinity or docking energy.
- Change made.
Reviewer 2 Report
The aim of this research was to develop new allosteric inhibitors using a virtual screening approach for dengue virus, a tropical disease with no existing treatment specific to it. The target protein for inhibition was NS2B/NS3 protease, a protein necessary for the dengue virus to replicate. The virtual screening approach used a thioguanine analogue designed by Hariono et al. as the starting point to find similar ligands via the Swiss Similarity online tool. Four compounds were identified that were stable in the allosteric binding site of the protease, and were also scored well for gastrointestinal absorption and low on crossing the blood brain barrier.
This virtual screening methodology is appropriate for the special issue regarding computer-aided drug design. The molecular docking, molecular dynamics, and free energy calculations combined provide a very detailed description of the ligand binding environment. However, the computational work only suggests potential avenues for future design, and would be supported by laboratory experiments, especially with regards to the toxicity and inhibition. In particular, the article by Hariono et al. describing the thioguanine analogue the authors used not only performed virtual screening to identify hit compounds, but they also synthesized in the lab 21 compounds based on the thioguanine scaffold. These compounds were then characterized by NMR and used in an inhibition assay for the protease.
The manuscript is well written, but there are a few minor typographical errors that should be corrected prior to publication. A few examples are below:
Line 50: "...three decades/ DENV has an..." should be "...three decades. DENV has an..."
Line 73: "...active site is plane..." should be "...active site is planar..."
Line 84: "no-competitive inhibitors" should be "non-competitive inhibitors"
Overall, this manuscript is well written, the calculations are clearly described and detailed, but the candidate compounds identified would still need to be experimentally verified in future research, for example, by the NS2B/NS3 protease inhibition assay.
Author Response
Point 02
The aim of this research was to develop new allosteric inhibitors using a virtual screening approach for dengue virus, a tropical disease with no existing treatment specific to it. The target protein for inhibition was NS2B/NS3 protease, a protein necessary for the dengue virus to replicate. The virtual screening approach used a thioguanine analogue designed by Hariono et al. as the starting point to find similar ligands via the Swiss Similarity online tool. Four compounds were identified that were stable in the allosteric binding site of the protease, and were also scored well for gastrointestinal absorption and low on crossing the blood brain barrier.
This virtual screening methodology is appropriate for the special issue regarding computer-aided drug design. The molecular docking, molecular dynamics, and free energy calculations combined provide a very detailed description of the ligand binding environment. However, the computational work only suggests potential avenues for future design, and would be supported by laboratory experiments, especially with regards to the toxicity and inhibition. In particular, the article by Hariono et al. describing the thioguanine analogue the authors used not only performed virtual screening to identify hit compounds, but they also synthesized in the lab 21 compounds based on the thioguanine scaffold. These compounds were then characterized by NMR and used in an inhibition assay for the protease.
The manuscript is well written, but there are a few minor typographical errors that should be corrected prior to publication. A few examples are below:
Line 50: "...three decades/ DENV has an..." should be "...three decades. DENV has an..."
Line 73: "...active site is plane..." should be "...active site is planar..."
Line 84: "no-competitive inhibitors" should be "non-competitive inhibitors"
Overall, this manuscript is well written, the calculations are clearly described and detailed, but the candidate compounds identified would still need to be experimentally verified in future research, for example, by the NS2B/NS3 protease inhibition assay.
Response 02
However, the computational work only suggests potential avenues for future design, and would be supported by laboratory experiments, especially with regards to the toxicity and inhibition.
Our research group develops theoretical works and, in addition, we have few resources for research in Brazil, which makes the feasibility of experimental validation impossible at the moment.
Line 50: "...three decades/ DENV has an..." should be "...three decades. DENV has an..."
- Change made.
Line 73: "...active site is plane..." should be "...active site is planar..."
- Change made.
Line 84: "no-competitive inhibitors" should be "non-competitive inhibitors"
- Change made.
Round 2
Reviewer 1 Report
Thank you for your responses.
The authors have responded satisfactorily to all the comments and made necessary changes to the manuscript.